# The Effects of Short-Term High-Intensity Interval Training and Moderate Intensity Continuous Training on Body Fat Percentage, Abdominal Circumference, BMI and VO_2max_ in Overweight Subjects

**DOI:** 10.3390/jfmk5020041

**Published:** 2020-06-10

**Authors:** Luca Russomando, Vincenzo Bono, Annamaria Mancini, Alessia Terracciano, Francesca Cozzolino, Esther Imperlini, Stefania Orrù, Andreina Alfieri, Pasqualina Buono

**Affiliations:** 1Dipartimento di Scienze Motorie e del Benessere, Università degli Studi di Napoli Parthenope, 80133 Naples, Italy; luca.russomando@uniparthenope.it (L.R.); vincenzo.bono@uniparthenope.it (V.B.); annamaria.mancini@uniparthenope.it (A.M.); alessia.terracciano@uniparthenope.it (A.T.); francesca.cozzolino@uniparthenope.it (F.C.); orru@uniparthenope.it (S.O.); 2CEINGE-Biotecnologie avanzate, 80131 Naples, Italy; 3IRCCS SDN, 80143 Naples, Italy; imperlini@ceinge.unina.it

**Keywords:** HIIT, MICT, VO_2max_, fat mass percentage

## Abstract

We aimed to compare the effects of a personalized short-term high-intensity interval training (HIIT) vs. standard moderate intensity continuous training (MICT) on body fat percentage, abdominal circumference, BMI and maximal oxygen uptake (VO_2max_) in overweight volunteers. Twenty overweight sedentary volunteers (24.9 ± 2.9y; BMI: 26.1 ± 1 kgm^−2^) were randomly assigned to 2 groups, HIIT or MICT. HIIT trained 6 weeks (3-days/week), 40-min sessions as follows: 6-min warm-up, 20-min resistance training (RT) at 70% 1-RM, 8-min HIIT up to 90% of the predicted Maximal Heart Rate (HR_max_), 6-min cool-down. MICT trained 6 weeks (3-days/week) 60-min sessions as follows: 6-min warm-up, 20-min RT at 70% 1-RM, 30-min MICT at 60–70% of the predicted HR_max_, 4-min cool-down. Two-way ANOVA was performed in order to compare the efficacy of HIIT and MICT protocols, and no significant interaction between training x time was evidenced (*p* > 0.05), indicating similar effects of both protocols on all parameters analyzed. Interestingly, the comparison of Δ mean percentage revealed an improvement in VO_2max_ (*p* = 0.05) together with a positive trend in the reduction of fat mass percentage (*p* = 0.06) in HIIT compared to MICT protocol. In conclusion, 6 weeks of personalized HIIT, with reduced training time (40 vs. 60 min)/session and volume of training/week, improved VO_2max_ and reduced fat mass percentage more effectively compared to MICT. These positive results encourage us to test this training in a larger population.

## 1. Introduction

Obesity represents one of the most important cardiovascular (CV) risk factors [1,2]. A lot of epidemiological and trial-based evidences suggest a direct link between body weight and health [3,4]. In Italy, more than one third of the adult population (35.5%) is overweight, whereas more than 10% of subjects is obese (10.4%): overall, 45.9% of people aged ≥18 years are overweight [5]. The American College of Sports Medicine (ACSM) in a recent position stand recommends aerobic physical activity 150 min/week for a modest weight loss (2–3 kg), 225–420 min/week for a considerable weight loss (5–7.5 kg), 200–300 min/week are indicated for the management of body weight after the weight loss [6]. However, due to lack of time, many people are often unable to reach the minimum level of recommended daily physical activity (PAL). According to ISTAT data, approximately 39.2% of Italian people do not attend any kind of sport and do not engage in physical activity in their free time [5]. In order to reduce sedentary lifestyle and to counteract overweight and obesity, research is recently focusing on the effectiveness of different short-term and high-intensity training protocols, that could represent a useful strategy for the management of body composition and the reduction of waist circumference [7]. In this context, MICTs are traditionally the most suitable protocols used for achieving these goals also in absence of food restriction in overweight subjects, and only recently, RT training has been included [8,9,10,11,12,13].

HIIT is considered a new-type of training, most popular among physically active individuals [14]: HIIT enhances 2–4% the performance in athletes improving both anaerobic and aerobic capacity [15,16,17,18]. More recently HIIT training has been proposed as an alternative protocol to MICT in order to improve physical fitness and body composition in healthy subjects [6,19]. Long-term (at least 12 weeks) High Intensity Interval Resistance Training (HIRT) improved resting energy expenditure and respiratory ratio to a greater extent than traditional MICT training also in non-dieting subjects [20].

Furthermore, the HIIT training is considered the first choice for healthy subjects who frequently dropped-out from gym training sessions due to lack of time. To date, there are no universally accepted guidelines reporting the length of the high-intensity session and the recovery time; a growing body of evidence suggests that HIIT protocols, i.e., 0.5–4 min bout of vigorous exercise interspersed by period of passive or active recovery, are a more-time efficient exercise strategy, also being perceived as more enjoyable in term of adherence [20,21,22,23].

Only few studies analyzed the effects of short-term HIIT protocols on body composition in different subjects, often providing conflicting results, so far [24,25,26]. Accordingly, the aim of the present study was to compare the effects of 6 weeks, 3 times/week, personalized supervised short-term HIIT (HIIT) vs. standard MICT protocol on BMI, body fat percentage, abdominal circumference and VO_2max_ in young overweight, sedentary subjects.

## 2. Materials and Methods

### 2.1. Study Design and Participant Recruitment

In this randomized control trial, young volunteers were randomly assigned to HIIT or MICT group to perform 6 weeks (3 days/week) of HIIT or MICT training, respectively. Anthropometry (height, weight, BMI), Body Fat mass percentage (%), abdominal circumference, together with VO_2max_ were evaluated at baseline (T0) and after 6 weeks (T1) of HIIT or MICT training protocols. Muscular fitness, 1-RM, was also evaluated at baseline, in all participants.

Twenty young, healthy, overweight, sedentary (performing less than 60 min, 1–2 times for week of exercise) subjects volunteered to participate in this study (24.9 ± 2.9y; height: 1.67 ± 0.1 (m); weight: 72.6 ± 9.1 (kg); BMI: 26.1 ± 1 (kg m^−2^); fat mass percentage: 25.8 ± 2.9 (%); abdominal circumference: 90.2 ± 5.8 cm; VO_2max_: 38.1 ± 2.2 (mL kg^−1^ min^−1^). The volunteers stated that they had a Mediterranean-type diet with no food-restrictions. The participants were asked to report any changes in lifestyle, including nutritional habits, or if they were taking any drugs during the study. Participants were randomly assigned to 2 different groups: 10 subjects (3 males and 7 females) to the HIIT group and 10 subjects (4 males and 6 females) to the MICT group, respectively. Participants were asked to refrain from consuming caffeine and alcohol and to not perform any physical activity for 24 h before each evaluation. Furthermore, in the seven days before the baseline assessment, participants completed a familiarization training session. Although ethics committee permission was not required prior to the beginning of the study [27], all participants were informed of the research aims and gave their written informed consent for participation in the study. The study was conducted nevertheless in accordance with the 1964 Helsinki Declaration and its later amendments or comparable ethical standards.

### 2.2. Measurements

#### 2.2.1. Anthropometric Evaluation

Height and weight were recorded using the SECA 756 (Mechanical Column Scale) with SECA 224 (Telescopic Measuring Rod for SECA column scales, respectively (SECA, Precision for health, 2017). The Body Fat percentage (%) was measured according to the Girth Method based on three girth measurements [28]; the measurement was approximately 0.1 cm.

Girth measurement sites are reported in Table 1; reference equations to predict Body Fat percentage are reported in Table 2.

##### Estimated VO_2max_

The estimation of VO_2max_ was performed using the single-stage treadmill Walking Test as described below [29]. Briefly, subjects first performed a 4-min warm-up at comfortable walking speed, based on the participant’s gender, age and fitness level, without slope, allowing the HR to reach the 50–70% of the predicted HR_max_. Then, participants continued at the same speed for additional 4 min at a 5% grade allowing to reach the Steady State HR (SS HR). The HR was recorded in the final 2 min, if the HR differed by more than 5 bpm, they continued the test for another 1 min and the new SS was recorded in the final 30-sec of the last 2 min. A cool down (2–5 min) was performed at the end of the test. The SS HR value will be used in the following equation to estimate the VO_2max_ (mL kg^−1^ min^−1^): 15.1 + (21.8 × speed in mph) – 0.327 (SS HR in bpm) – 0.263 (speed × age in y) + 0.00504 (SS HR in bpm × age in y) + 5.98 (gender: female = 0, male = 1) [29].

#### 2.2.2. 1-RM Estimation

1-RM evaluation was estimated by Brzycki’ s regression equation, 1-RM = Kg/(1.0278 − (RM × 0.0278)) [30].

### 2.3. RT Protocol

All participants performed a 6-min warm-up and 20 min of the RT 3 times per week for 6 weeks as follows: 3 sets of 12 reps (at ~70% 1-RM); 1-min rest periods between sets. The RT training was performed using isotonic machines, i.e., leg press, lat machine, chest press, crunch machine. After 20 min of RT protocol, the subjects performed HIIT or MICT training protocol, respectively.

### 2.4. HIIT Protocol

A YMCA adapted fitness protocol [31] was used in order to determine the work rate corresponding to the 85% of HR_max_ for each recruited subject. Briefly, all participants performed the 1st stage, 3 min at a work rate of 150 kg/min. According to the HR measured at the end of the 1st stage, indicated in the rows of Table 3, subjects continued the YMCA protocol by performing the 2nd up to the 5th (if required) stages (each of 3 min) at the workload reported in the corresponding columns (see Table 3). HR was measured during the final 15–30 s of the second and third minute; the work rate continued for an additional minute if the HR values varied more than 5 beats/min between the second and the third minute. The protocol was interrupted when the HR reached the 85% of the predicted HR_max_. HR_max_ was evaluated by Tanaka equation: 208 − (0.7 × age) [32]. At the end of the test, the workload (kgm/min) and HR were recorded, representing the starting point for the personalized HIIT setting protocol.

A Jump box (SportPlus, PLYO, Hamburg, Germany) with adjustable height from 31 to 51 cm for the high-intensity phases and a stepper (Mirafit Stepper deluxe, Norfolk, UK) with height adjustable from 10 to 30 cm for recovery phases were used, respectively.

Starting from the results of YMCA test, HIIT protocol was personalized as follows: 4 steps, 120 s each, composed of 30 s at 140% of the maximum workload (kgm/min) leading to the increase of HR up to 90% of the predicted HR_max_ and 90 s of active recovery performed at 25% of the maximum workload (kgm/min) allowing to reduce the HR up to 60% of the predicted HR_max_. The intensity (kgm/min) was increased by 5% every 2 weeks.

### 2.5. MICT Protocol

After the RT section, the MICT protocol proceeds for additional 30 min with a continuous cardiovascular training session performed at 60–70% of the predicted HR_max_, using a treadmill.

### 2.6. Data Analysis

Two-way ANOVA was carried-out to compare the effects of HIIT and MICT after the training (T1) on BMI, Fat mass percentage, Abdominal Circumference and VO_2max_, vs. baseline (T0) in all subjects. Delta (Δ) values percentage were obtained calculating the percentage change at T1 vs. T0 as follows (T1/T0 × 100) − 100 percent. Then the mean Δ values obtained for MICT and HIIT were compared using one-way ANOVA; *p*-value ≤ 0.05 was considered significant. Data were reported as mean ± SE; data were analyzed using Statview statistical software (version 5.0.1.0; SAS Institute).

## 3. Results

Seventeen subjects completed the training program; three subjects (1 male and 2 females) belonging to MICT group dropped-out after 2 weeks of sessions. None of the recruited subjects reported any training-related injury.

The HIIT and MICT groups did not differ significantly at baseline either in anthropometric characteristics or in VO_2max_ (Table 4). Subjects belonging to the HIIT or the MICT group participated in 17.6 ± 0.5 and 17.1 ± 0.7 sessions, respectively (*p* = 0.103), on a total of 18 scheduled sessions.

In order to compare the effects of HIIT vs. MICT protocol on body composition and VO_2max_ in overweight volunteers, we conducted a two-way ANOVA with repeated measures (Figure 1, A1–A4) that revealed improvements in all tested parameters, irrespective to the protocol used. Moreover, two-way ANOVA did not show a significant interaction between training and time (training × time interaction, *p* > 0.05 for all parameters analyzed), indicating that both training protocols provided similar positive effects on the variables analyzed.

Furthermore, to verify if the two protocols gave different performances, we compared the (Δ, T1−T0) mean percentage for each parameter between HIIT and MICT using the one-way ANOVA (see Table 5). A more effective improvement in the VO_2max_ (*p* = 0.05) together with a positive trend in fat mass percentage reduction (*p* = 0.06) in HIIT compared to MICT protocol, was evidenced.

## 4. Discussion

The main goal of the present study was to compare the effect of a short-term supervised personalized high-intensity interval training (HIIT) vs. standard moderate intensity continuous training (MICT) on body fat percentage, abdominal circumference, BMI and maximal oxygen uptake (VO_2max_) in young overweight volunteers. In summary, we demonstrated that 6 weeks of HIIT, with reduced training time (40 vs. 60 min)/session and volume of training/week improved VO_2max_ and reduced fat mass percentage more effectively compared to MICT.

The greater VO_2max_ improvement in HIIT group vs. MICT suggests that exercise intensity, more than volume, is a critical aspect of training adaptations [33]. Our findings are in line with previous studies reporting that HIIT training led to a greater improvement in VO_2max_ when compared to traditional circuit training [34] and similarly to that obtained with MICT protocols [35,36,37].

Few and conflicting results on the positive effects of HIIT on the reduction of Body Fat percentage compared to standard MICT protocols have been provided [33,38,39]; other studies showed no significant reduction in Body Fat percentage or improvement in the fat distribution after HIIT protocols in overweight subjects [25,26,33]. Furthermore, a recent meta-analysis, investigating the efficacy of HIIT compared to MICT training protocol on body composition improvement, highlighted that HIIT protocol reduced fat mass percentage, similarly to that obtained using MICT protocol [40].

Finally, the length of intervention time (weeks) is an important factor correlated to the effectiveness of HIIT protocol: in a meta-analysis study, in fact, Batacan et al. [21] assessed that at least 12 weeks of HIIT protocol are needed to achieve significant improvement in cardio-metabolic parameters and in body composition both in normal-weight and obese subjects. Here, we suggested that 6 weeks of personalized HIIT protocol are enough to induce a more effective improvement in VO_2max_ compared to MICT.

Furthermore, our results contribute to the growing evidence reporting that HIIT training turns to be as effective but more time-efficient when compared to MICT protocols [41]. Such newer elements are important given that they are the most commonly existing drawback to regular exercise participation [42].

A limit of the study is the small number of subjects investigated, but the positive results encourage us to test this training in a larger population. Another aspect that should be evaluated is the adherence to the HIIT protocols; in fact, to date, there is still debate about the adherence level to HIIT protocols in different studies [43,44]. Further, the association between HIIT and compliance should represent the starting point of future studies aimed to identify the minimum effective dose of HIIT to improve health status. Such efforts would be important to increase the public health impact of HIITs.

## 5. Conclusions

Our findings suggest that 6 weeks of personalized HIIT protocol (40 min/session) represent a valid alternative to the standard moderate physical activity training (MICT) (60min/session) in order to improve health-related parameters: BMI, fat mass percentage, abdominal circumference and VO_2max_ in overweight young sedentary subjects. We demonstrated that 6 weeks of personalized HIIT, with reduced training time (40 vs. 60 min)/session and volume of training/week, improved VO_2max_ and reduced fat mass percentage more effectively compared to MICT. A limit of this study is the low number of participants. The positive results encourage us to test this training in a larger population including non-communicable disease patients, in order to better understand the effectiveness of this personalized HIIT protocol on cardio-metabolic parameters and its compliance.

## Figures and Tables

**Figure 1 jfmk-05-00041-f001:**
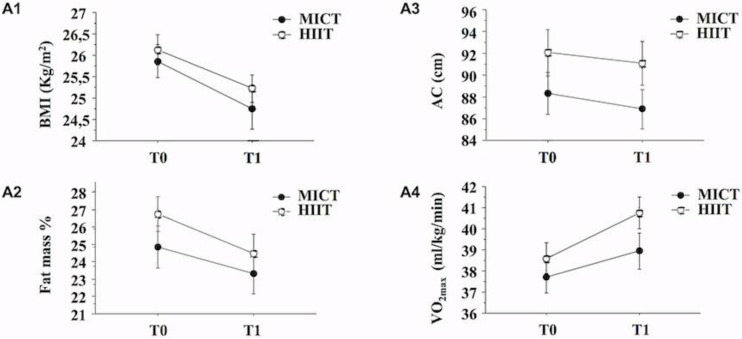
Effects of moderate intensity continuous training (MICT) and HIIT protocol on BMI, Abdominal Circumference (AC), Fat mass percentage and VO_2max_. (A1–A4): Line charts report two-way ANOVA with repeated measures at baseline (T0) and after 6 weeks of training (T1) with BMI (**A1**), Body Fat mass percentage (**A2**), Abdominal circumference, AC (**A3**) and VO_2max_ (**A4**) as dependent variables. All values are expressed as the mean ± SE. No significant interaction (*p* > 0.05) between training and time (training × time interaction) for all parameters analyzed were found.

**Table 1 jfmk-05-00041-t001:** Girth measurement sites according to age and gender.

**Age (y)**	**Gender**	**Site A**	**Site B**	**Site C**
18–26	Males	Right upper arm	Abdomen	Right forearm
18–26	Females	Abdomen	Right thigh	Right forearm
27–50	Males	Buttocks	Abdomen	Right forearm
27–50	Females	Abdomen	Right thigh	Right calf

**Table 2 jfmk-05-00041-t002:** Age and gender-specific equations to predict body fat percentage.

**Age** (**y**)	**Gender**	**Equations**
18–26	Males	Constant A + Constant B − Constant C − 10.2
18–26	Females	Constant A + Constant B − Constant C – 19.6
27–50	Males	Constant A + Constant B − Constant C – 15
27–50	Females	Constant A + Constant B − Constant C – 19.6

**Table 3 jfmk-05-00041-t003:** YMCA adapted fitness protocol used for setting personalized HIIT protocol.

**1st Stage 150 kgm/min for 3 min**				
	**HR: <80 bpm**	**HR: 80-89 bpm**	**HR: 90-100 bpm**	**HR: >100 bpm**
**2nd stage**	750 kgm/min	600 kgm/min	450 kgm/min	300 kgm/min
**3rd stage**	900 kgm/min	750 kgm/min	600 kgm/min	450 kgm/min
**4th stage**	1050 kgm/min	900 kgm/min	750 kgm/min	600 kgm/min
**5th stage**	1200 kgm/min	1050 kgm/min	900 kgm/min	750 kgm/min

**Table 4 jfmk-05-00041-t004:** Baseline characteristics of the two groups of volunteers participating in HIIT or MICT protocols.

**Parameters**	***HIIT***	***MICT***
Gender (M/F)	3M/7F	4M/6F
Age (y)	24 ± 3	26 ± 2
Height (m)	1.65 ± 0.12	1.68 ± 0.09
Weight (kg)	71.6 ± 10.9	73.7 ± 7.4
BMI (kg m^−2^)	26.1 ± 1.1	25.9 ± 0.9
Body Fat percentage (%)	26.3 ± 3	25.2 ± 2.9
Abdominal circumference (cm)	92.1 ± 6.7	88.2 ± 4.3
VO_2max_ (mL kg^−1^ min^−1^)	38.6 ± 2.3	37.6 ± 2.1

Values were represented as mean ± SD. No significant differences between groups at baseline were evidenced. BMI= Body Mass Index.

**Table 5 jfmk-05-00041-t005:** Comparison of Δ (T1−T0) mean percentage between MICT and HIIT for BMI, Fat mass %, Abdominal Circumference and VO_2max_. Analysis was performed using one-way ANOVA; for each parameter, we indicated the mean percentage (Δ Mean percentage), the Standard Error (SE) and *p* value. * *p*-value ≤ 0.05.

	**Training**	**Δ Mean Percentage**	**SE**	***p*** **Value**
BMI (Kg/m^2^)	MICT HIIT	−4.3 −3.4	0.9 0.3	0.28
Fat mass %	MICT HIIT	−5.7 −8.2	0.7 0.9	0.06
Abdominal Circumference (cm)	MICT HIIT	−1.6 −1.1	0.3 0.4	0.30
VO_2max_ (mL/Kg/min)	MICT HIIT	+3.2 +5.7	0.5 0.9	0.05 *

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
