# Peer review of "The Effects of Short-Term High-Intensity Interval Training and Moderate Intensity Continuous Training on Body Fat Percentage, Abdominal Circumference, BMI and VO2max in Overweight Subjects"

_jfmk, 2020, doi:10.3390/jfmk5020041_

Round 1
Reviewer 1 Report
Dear Authors
the topic is of interest and I found just a few minors to be solved.
The introduction is well designed and clear, but I suggest to add a few references regarding the importance of High-Intensity Interval Resistance Training in Non-Dieting Individuals and also the impact that fitness activities have on body composition of people. The methods section is logically written and clear. Also, the results are informative and well presented. The last part of the manuscript is lacking a bit of study limitation (appears sometimes overpromising). I suggest editing, accordingly.
In general, it is a good manuscript.
Author Response
Reviewer #1
Dear Authors
the topic is of interest and I found just a few minors to be solved.
The introduction is well designed and clear, but I suggest to add a few references regarding the importance of High-Intensity Interval Resistance Training in Non-Dieting Individuals and also the impact that fitness activities have on body composition of people. The methods section is logically written and clear. Also, the results are informative and well presented. The last part of the manuscript is lacking a bit of study limitation (appears sometimes overpromising). I suggest editing, accordingly.
In general, it is a good manuscript.
Thanks for your suggestions: now we edited the Introduction and the Discussion section, accordingly.
Reviewer 2 Report
The study by Russomando et al. compared changes in weight and fitness after 6 weeks of HIIT vs MICT in healthy volunteers.
Suggest removing the a in front of HIIT. I am not sure what the adapted part means, but there are many different adapted HIIT protocols – as long as alternates bout of high intensity ‘near max’ with low intensity it is HIIT.
Suggest changing to ‘estimated VO2peak’ this test does not meet the criterial for measuring VO2max. It is an estimation for peak.
First sentence of introduction = poorly written. ‘was due’ you mean is characterized/classified? Regardless, I wouldn’t think you need to define obesity.
Line 69 – repeat use of ‘to date’ suggest using another term
Line 74 – add in if was free-living at home exercise or supervised
Line 85 – ‘they showed a similar diet recall’ Should be in results with data. Instead explain how and when the recall was collected.
Please clearly explain the HIIT protocol. From what I can understand they did 3 min intervals of increasing HR but I don’t understand the table it has 4 levels that go across in the rows and then 4 columns down of 3 min intervals.
Statistics – so you compared the change from pre-post between HIIT and MICT?
Otherwise it’s a two way / repeated measures ANOVA pre and post for HIIT and MICT.
You cannot separately compare the change for HIIT and then separately for MICT, unless you have an interaction on the ANOVA.
This will tell you if HIIT is better than MICT – which is/should be main question?
The graphs and results are good, very tight. But we already know exercise improves these, what we want to know is if your modified HIIT improves more/similar to MICT – therefore the graphs should show the change for HIIT vs MICT.
Here, Figure 3. Should be a change from T0
Discussion – it may be personal preference but I like the discussion to start with what YOUR main findings were and how these contribute to knowledge gain.
Author Response
Reviewer #2
The study by Russomando et al. compared changes in weight and fitness after 6 weeks of HIIT vs MICT in healthy volunteers.
Suggest removing the a in front of HIIT. I am not sure what the adapted part means, but there are many different adapted HIIT protocols – as long as alternates bout of high intensity ‘near max’ with low intensity it is HIIT.
Thank you for your comment; we removed the “a” in front of HIIT everywhere in the manuscript, accordingly. We substituted “a” with “s” that means short-term; now we read sHIIT alongside the paper.
We also deleted the term “adapted” from the title; now we read : “ The effects of short term high intensity interval training and moderate intensity continuous training on body fat percentage, abdominal circumference, BMI and VO2max in overweight subjects”
Suggest changing to ‘estimated VO2peak’ this test does not meet the criterial for measuring VO2max. It is an estimation for peak.
Thank you for the comment. We used the Single-stage treadmill Walking Test (Ebelling, 1991) submaximal in order to estimate the VO2max by using appropriate equation. Further, considering that the bpm differences between the last 2 min of the test should be less than 5 bpm, it is concevaible that estimated VO2max equal to estimated VO2peack. So, we referred to the VO2max estimation in the paper.
First sentence of introduction = poorly written. ‘was due’ you mean is characterized/classified? Regardless, I wouldn’t think you need to define obesity.
Thank you for your comment; we deleted the first sentence of introduction, accordingly.
Line 69 – repeat use of ‘to date’ suggest using another term
We changed the term “to date” as suggested; now we read, “Only few studies analyzed the effects of short-term HIIT protocols on body composition in different group of subjects, often providing conflicting results, so far”.
Line 74 – add in if was free-living at home exercise or supervised
We added “supervised” as requested.
Line 85 – ‘they showed a similar diet recall’ Should be in results with data. Instead explain how and when the recall was collected.
Thank you for your observation. We edited in the 2.1.” Study design and participant recruitment” section, accordingly.
- Please clearly explain the HIIT protocol. From what I can understand they did 3 min intervals of increasing HR but I don’t understand the table it has 4 levels that go across in the rows and then 4 columns down of 3 min intervals.
Thanks for your suggestions; now, we described the HIIT protocol in greater details in the method section; we also modified table 3, accordingly.
Statistics – so you compared the change from pre-post between HIIT and MICT? Otherwise it’s a two way / repeated measures ANOVA pre and post for HIIT and MICT. You cannot separately compare the change for HIIT and then separately for MICT, unless you have an interaction on the ANOVA.
This will tell you if HIIT is better than MICT – which is/should be main question? The graphs and results are good, very tight. But we already know exercise improves these, what we want to know is if your modified HIIT improves more/similar to MICT – therefore the graphs should show the change for HIIT vs MICT.
According to your observations, we performed a Two-Way ANOVA with repeated measures. Results are now reported in the manuscript (see new figure 1). Next, we compared the Delta mean (T1-T0) of each parameter between MICT and sHIIT protocol and the results are now reported in the new table 5.
Discussion – it may be personal preference but I like the discussion to start with what YOUR main findings were and how these contribute to knowledge gain.
Thanks for your suggestion. We edited the discussion section, accordingly.
Reviewer 3 Report
The manuscript by Russomando and colleagues aims to open solutions to overcome time-related obstacles to exercise, suggesting short-term high intensity interval training rather than moderate intensity continuous training in order to achieve similar outcomes by overweight young subjects.
Overall, the manuscript is well written and concise. The aim is simple and clearly stated, the introduction sufficientely explains the reasons behind the study, the methods are appropriate, the results need some corrections in data processing and the discussion is clear.
There is value in publishing this paper, so I recommend the acceptance.
Few changes/corrections could, although, improve this paper:
- Lines 63-68: this paragraph seems to follow a chronological definition of HIIT when, instead, the three cited papers were published exactly in reverse order. Provide a more conceptual, rather than chronological, meaning to this description.
- Figure 1 and 2: in order to provide information about the single values and describe the distribution, it is more appropriate to use the box or dot plots rather than the line charts.
- Figure 3: it is not correct to estimate the difference in effectiveness between aHIIT and MICT by comparing, for each parameter, the respectively T1 values, since they are only indicative of the final measurements. Differences between the two protocols will be outlined when comparing the change/delta of each parameter (percentage difference between baseline (T0) and final assessment (T1) values), or by data normalization.
- Lines 219-220/225/232-233, in discussion and conclusion, refer to an association between aHIIT and better compliance by participants. The results of this study cannot allow to state that conclusion. Beyond the fact that the authors did not investigate and analyze the reasons behind the drop-out, the entire sample was too small to consider 3 people as a statistically significant value to draw conclusions on program adherence.
Author Response
Reviewer #3
The manuscript by Russomando and colleagues aims to open solutions to overcome time-related obstacles to exercise, suggesting short-term high intensity interval training rather than moderate intensity continuous training in order to achieve similar outcomes by overweight young subjects.
Overall, the manuscript is well written and concise. The aim is simple and clearly stated, the introduction sufficiently explains the reasons behind the study, the methods are appropriate, the results need some corrections in data processing and the discussion is clear.
There is value in publishing this paper, so I recommend the acceptance.
Few changes/corrections could, although, improve this paper:
- Lines 63-68: this paragraph seems to follow a chronological definition of HIIT when, instead, the three cited papers were published exactly in reverse order. Provide a more conceptual, rather than chronological, meaning to this description.
Thank you for your suggestions; now, we edited in introduction paragraph, accordingly.
-Figure 1 and 2: in order to provide information about the single values and describe the distribution, it is more appropriate to use the box or dot plots rather than the line charts.
Thank you for your suggestions, moreover, we re-wrote part of the results by focusing on the main objective of the manuscript: the comparison between the two protocols. So, we removed figures 1 and 2 and we compared the effects of sHIIT and MICT protocols on Body composition and VO2max by the two-way ANOVA with repeated measures. Results are now shown in the new Figure 1.
- Figure 3: it is not correct to estimate the difference in effectiveness between aHIIT and MICT by comparing, for each parameter, the respectively T1 values, since they are only indicative of the final measurements. Differences between the two protocols will be outlined when comparing the change/delta of each parameter (percentage difference between baseline (T0) and final assessment (T1) values), or by data normalization.
Thank you for your suggestions; now we compared the Delta mean (T1-T0) for each parameter between MICT and sHIIT protocol using the one-way ANOVA test. Results are reported in the new table 5.
- Lines 219-220/225/232-233, in discussion and conclusion, refer to an association between aHIIT and better compliance by participants. The results of this study cannot allow to state that conclusion. Beyond the fact that the authors did not investigate and analyze the reasons behind the drop-out, the entire sample was too small to consider 3 people as a statistically significant value to draw conclusions on program adherence.
Thanks for your suggestions: we edited the discussion and conclusion sections, accordingly.
Round 2
Reviewer 2 Report
Author's Notes
Reviewer #2
The study by Russomando et al. compared changes in weight and fitness after 6 weeks of HIIT vs MICT in healthy volunteers.
Suggest removing the a in front of HIIT. I am not sure what the adapted part means, but there are many different adapted HIIT protocols – as long as alternates bout of high intensity ‘near max’ with low intensity it is HIIT.
Thank you for your comment; we removed the “a” in front of HIIT everywhere in the manuscript, accordingly. We substituted “a” with “s” that means short-term; now we read sHIIT alongside the paper.
We also deleted the term “adapted” from the title; now we read : “ The effects of short term high intensity interval training and moderate intensity continuous training on body fat percentage, abdominal circumference, BMI and VO2max in overweight subjects”
Nice, like the title change. Would still suggest having nothing before HIIT? I don’t feel it is needed.
Suggest changing to ‘estimated VO2peak’ this test does not meet the criterial for measuring VO2max. It is an estimation for peak.
Thank you for the comment. We used the Single-stage treadmill Walking Test (Ebelling, 1991) submaximal in order to estimate the VO2max by using appropriate equation. Further, considering that the bpm differences between the last 2 min of the test should be less than 5 bpm, it is concevaible that estimated VO2max equal to estimated VO2peack. So, we referred to the VO2max estimation in the paper.
First sentence of introduction = poorly written. ‘was due’ you mean is characterized/classified? Regardless, I wouldn’t think you need to define obesity.
Thank you for your comment; we deleted the first sentence of introduction, accordingly.
Please remove actually.
Line 69 – repeat use of ‘to date’ suggest using another term
We changed the term “to date” as suggested; now we read, “Only few studies analyzed the effects of short-term HIIT protocols on body composition in different group of subjects, often providing conflicting results, so far”.
Line 74 – add in if was free-living at home exercise or supervised
We added “supervised” as requested.
Line 85 – ‘they showed a similar diet recall’ Should be in results with data. Instead explain how and when the recall was collected.
Thank you for your observation. We edited in the 2.1.” Study design and participant recruitment” section, accordingly.
- Please clearly explain the HIIT protocol. From what I can understand they did 3 min intervals of increasing HR but I don’t understand the table it has 4 levels that go across in the rows and then 4 columns down of 3 min intervals.
Thanks for your suggestions; now, we described the HIIT protocol in greater details in the method section; we also modified table 3, accordingly.
The HIIT protocol is still extremely hard to follow. I am still so confused…..
So it was stepping up 31-51 cm for high intensity and 10cm for recovery intervals. They did 4x 2-min intervals that increased in intensity as per table – is it across or down?.
What is the 4x 3-min in the table?
Important points to clarify:
The workload to elicit an intensity of xxxx for the intervals was set individually using a preliminary step test.
- Line 130 What was the HR value you were aiming for i.e., intensity HIIT intervals is ~90% of maximal HR
- e., what was the HR level you were aiming for to base that level on
Statistics – so you compared the change from pre-post between HIIT and MICT? Otherwise it’s a two way / repeated measures ANOVA pre and post for HIIT and MICT. You cannot separately compare the change for HIIT and then separately for MICT, unless you have an interaction on the ANOVA.
This will tell you if HIIT is better than MICT – which is/should be main question? The graphs and results are good, very tight. But we already know exercise improves these, what we want to know is if your modified HIIT improves more/similar to MICT – therefore the graphs should show the change for HIIT vs MICT.
According to your observations, we performed a Two-Way ANOVA with repeated measures. Results are now reported in the manuscript (see new figure 1). Next, we compared the Delta mean (T1-T0) of each parameter between MICT and sHIIT protocol and the results are now reported in the new table 5.
Thank you for clarifying.
Table 5 and Figure one show the same results, and therefore table is not needed.
Discussion – it may be personal preference but I like the discussion to start with what YOUR main findings were and how these contribute to knowledge gain.
Thanks for your suggestion. We edited the discussion section, accordingly.

Author Response
Journal: JFMK (ISSN 2411-5142)
Manuscript ID: jfmk- 785469
Point by point’s response (in green):
Reviewer #2
The study by Russomando et al. compared changes in weight and fitness after 6 weeks of HIIT vs MICT in healthy volunteers.
Suggest removing the a in front of HIIT. I am not sure what the adapted part means, but there are many different adapted HIIT protocols – as long as alternates bout of high intensity ‘near max’ with low intensity it is HIIT.
Thank you for your comment; we removed the “a” in front of HIIT everywhere in the manuscript, accordingly. We substituted “a” with “s” that means short-term; now we read sHIIT alongside the paper.
We also deleted the term “adapted” from the title; now we read : “ The effects of short term high intensity interval training and moderate intensity continuous training on body fat percentage, abdominal circumference, BMI and VO2max in overweight subjects”
Nice, like the title change. Would still suggest having nothing before HIIT? I don’t feel it is needed.
Thank you for your comments. We also removed the “s” in front of HIIT, everywhere in the manuscript, accordingly.
Suggest changing to ‘estimated VO2peak’ this test does not meet the criterial for measuring VO2max. It is an estimation for peak.
Thank you for the comment. We used the Single-stage treadmill Walking Test (Ebelling, 1991) submaximal in order to estimate the VO2max by using appropriate equation. Further, considering that the bpm differences between the last 2 min of the test should be less than 5 bpm, it is concevaible that estimated VO2max equal to estimated VO2peack. So, we referred to the VO2max estimation in the paper.
First sentence of introduction = poorly written. ‘was due’ you mean is characterized/classified? Regardless, I wouldn’t think you need to define obesity.
Thank you for your comment; we deleted the first sentence of introduction, accordingly.
Please remove actually.
We deleted “actually” accordingly.
Line 69 – repeat use of ‘to date’ suggest using another term
We changed the term “to date” as suggested; now we read, “Only few studies analyzed the effects of short-term HIIT protocols on body composition in different group of subjects, often providing conflicting results, so far”.
Line 74 – add in if was free-living at home exercise or supervised
We added “supervised” as requested.
Line 85 – ‘they showed a similar diet recall’ Should be in results with data. Instead explain how and when the recall was collected.
Thank you for your observation. We edited in the 2.1.” Study design and participant recruitment” section, accordingly.
Please clearly explain the HIIT protocol. From what I can understand they did 3 min intervals of increasing HR but I don’t understand the table it has 4 levels that go across in the rows and then 4 columns down of 3 min intervals.
Thanks for your suggestions; now, we described the HIIT protocol in greater details in the method section; we also modified table 3, accordingly.
The HIIT protocol is still extremely hard to follow. I am still so confused…..
So it was stepping up 31-51 cm for high intensity and 10cm for recovery intervals. They did 4x 2-min intervals that increased in intensity as per table – is it across or down?.
What is the 4x 3-min in the table?
Important points to clarify:
The workload to elicit an intensity of xxxx for the intervals was set individually using a preliminary step test.
- Line 130 What was the HR value you were aiming for i.e., intensity HIIT intervals is ~90% of maximal HR
o i.e., what was the HR level you were aiming for to base that level on
According to your observations, we added even more details in the protocol description, hoping it will be more comprehensive now; we also modified the table 3.
- Line 130: We detailed the sentence, accordingly.
Statistics – so you compared the change from pre-post between HIIT and MICT? Otherwise it’s a two way / repeated measures ANOVA pre and post for HIIT and MICT. You cannot separately compare the change for HIIT and then separately for MICT, unless you have an interaction on the ANOVA.
This will tell you if HIIT is better than MICT – which is/should be main question? The graphs and results are good, very tight. But we already know exercise improves these, what we want to know is if your modified HIIT improves more/similar to MICT – therefore the graphs should show the change for HIIT vs MICT.
According to your observations, we performed a Two-Way ANOVA with repeated measures. Results are now reported in the manuscript (see new figure 1). Next, we compared the Delta mean (T1-T0) of each parameter between MICT and sHIIT protocol and the results are now reported in the new table 5.
Thank you for clarifying.
Table 5 and Figure one show the same results, and therefore table is not needed.
Table 5 and Figure 1 show different results. In particular, Table 5 shows the comparison of D (T1-T0) mean percentage between MICT and HIIT protocols for the different parameters by one-way ANOVA. This approach excluded the bias due to the initial differences of the recruited subjects and allowed us to evidence a more effective improvement in V02max in HIIT compared to MICT protocol. Figure 1 reported the two-way ANOVA analysis, indicating that both training protocols worked in the same way, providing similar improvements on body composition and V02max.
We’ll better specify it in our results.
Discussion – it may be personal preference but I like the discussion to start with what YOUR main findings were and how these contribute to knowledge gain.
Thanks for your suggestion. We edited the discussion section, accordingly.

Round 3
Reviewer 2 Report
thank you for making the suggested changes